# High-Yield-Related Genes Participate in Mushroom Production

**DOI:** 10.3390/jof10110767

**Published:** 2024-11-05

**Authors:** Fang Wang, Fengzhu Li, Luyang Han, Jingzi Wang, Xupo Ding, Qinhong Liu, Mingguo Jiang, Hailin Li

**Affiliations:** 1Guangxi Key Laboratory of Polysaccharide Materials and Modification, School of Marine Sciences and Biotechnology, Guangxi Minzu University, Nanning 530008, China; wf@gxmzu.edu.cn (F.W.); 19533366290@163.com (F.L.); reyarhan@163.com (L.H.); wangjinzi@gxun.edu.cn (J.W.); xupoding@hotmail.com (X.D.); 2Department of Vegetables, College of Horticulture, China Agricultural University, Beijing 100193, China; qhliu@cau.edu.cn

**Keywords:** mushroom, high yield, breeding, genes

## Abstract

In recent years, the increasing global demand for mushrooms has made the enhancement of mushroom yield a focal point of research. Currently, the primary methods for developing high-yield mushroom varieties include mutation- and hybridization-based breeding. However, due to the long breeding cycles and low predictability associated with these approaches, they no longer meet the demands for high-yield and high-quality varieties in the expansive mushroom market. Modern molecular biology technologies such as RNA interference (RNAi) and gene editing, including via CRISPR-Cas9, can be used to precisely modify target genes, providing a new solution for mushroom breeding. The high-yield genes of mushrooms can be divided into four categories based on existing research results: the genes controlling mycelial growth are very suitable for genetic modification; the genes controlling primordium formation are directly or indirectly regulated by the genes controlling mycelial growth; the genes controlling button germination are more difficult to modify; and the genes controlling fruiting body development can be regulated during the mycelial stage. This article reviews the current research status for the four major categories of high-yield-related genes across the different stages of mushroom growth stages, providing a foundation and scientific basis for using molecular biology to improve mushroom yield and promote the economic development of the global edible-mushroom industry.

## 1. Introduction

Edible mushrooms are recommended by the United Nations Food and Agriculture Organization (FAO) as one of the three cornerstones of a healthy diet, alongside meat and vegetables, due to their distinctive flavor and nutritional richness, which makes them highly appealing to consumers [1,2,3,4,5,6,7,8,9,10,11]. According to the latest statistics from the FAO, the annual growth rate of global mushroom production exceeds 8%. From 2010–2020, mushroom production increased from 24.977 million tons to 42.7929 million tons, representing a growth of 71.33% [12]. The global demand for mushrooms is substantial, leading many countries to establish comprehensive industrial chains related to mushroom production. With the rapid expansion of the mushroom market, the demand for high-quality and high-yield strains has surged. However, several challenges persist in mushroom cultivation, such as strain degradation [13,14,15,16,17,18,19,20,21]. However, the complex genetic background of mushrooms, along with the low-efficiency, time-consuming nature, and labor-intensive characteristics of traditional breeding methods, as well as the highly unstable breeding trajectories, hinder effective improvement [22,23,24,25]. For instance, in a preliminary study examining the mutagenic effects of ^60^Co-γ rays on basidiospores of *Agaricus bisporus*, the spore mortality rate under six different irradiation doses ranged from 50.6 to 90.63%, with only 0.76% maintaining the parental colony characteristics. Additionally, in the cultivation of superior strains of *Hypsizygus marmoreus* through crossbreeding technology, it is necessary to pair 94 spore monokaryons to conduct routine mating type identification, requiring a total of 8836 pairings; this extensive workload significantly limits the feasibility of the method [26,27]. Consequently, the development of a stable and efficient molecular breeding technology is a crucial foundation for both basic research on and industrial advancement in mushroom cultivation. Currently, the use of molecular biology markers to support hybrid breeding technology offers distinct advantages compared to traditional breeding methods. For example, the new *Agaricus bisporus* cultivar Fumo 48, under optimal cultivation conditions, demonstrates average yields comparable to the prominent foreign cultivar A15, while exceeding those of As2796 by 20–25%. This cultivar is well suited for year-round modern farm cultivation and appropriate for fresh market sales [28]. Nevertheless, the broader promotion and application of other molecular biology techniques in mushroom breeding necessitate the establishment of a mature and secure technical framework.

Modern molecular biology techniques, including omics, RNA interference (RNAi), and gene editing—particularly the novel CRISPR-Cas 9 technology—enable precise identification and editing of target genes, facilitating the targeted enhancement of high-yield characteristics in mushrooms [29,30,31,32,33]. Through omics research, it is possible to identify target genes that regulate high-yield characteristics in mushrooms. This can be followed by precise screening of high-yield-related genes using advanced artificial intelligence (AI) technologies. The rapid processing, analysis, and precise targeting capabilities of AI make it particularly well suited for the screening of high-yield genes [34,35,36,37]. Currently, the fusion of biology and artificial intelligence has been extensively applied in the field of synthetic biology, resulting in the establishment of relevant technology platforms and databases. However, the integration of molecular biology and artificial intelligence still requires further investigation and research for effective application [38].

After screening the high-yield-related genes, RNAi technology can be employed to systematically investigate the function of the target high-yield-related genes. Its technical advantages include rapid action, high stability, relative ease of operation, and the capability to target multiple genes simultaneously in a single procedure [39,40,41]. Once the roles of these genes in increasing yield enhancement are clarified, the genes can be overexpressed in mushroom cells using gene editing and CRISPR-Cas9 technology, facilitating the development of new high-yield and high-quality strains. The technical advantages of gene editing include convenient operation, low cost, and high efficiency of genetic transformation mediated by *Agrobacterium tumefaciens*. Additionally, CRISPR-Cas9 technology allows for efficient and rapid targeted editing at specific genomic loci in the genome, enabling knockout, insertion, and repair functionalities [42,43,44]. Taking the research on multiplex gene precise editing and large DNA fragment deletion using the CRISPR-Cas9-TRAMA system in edible mushroom *Cordyceps militaris* as an example, the synthetases of cordycepin and ergothioneine possess functional characteristics that inhibit cancer and skin aging. Employing CRISPR-Cas9 technology to investigate the functional characteristics of the cordycepin and ergothioneine genes, while simultaneously enhancing the yield of *Cordyceps militaris*, could significantly promote the economic benefits for both the pharmaceutical and cosmetic industries [45]. Consequently, the application of modern molecular biology techniques for high-yield strains is anticipated to address the challenges associated with developing high-yield mushroom varieties, providing a novel approach to achieving industrial breakthroughs and overcoming existing bottlenecks [46,47,48,49]. Although molecular biology technology has significantly shortened the breeding cycle of mushrooms and accelerated the research and development of new varieties, challenges remain regarding safety, social acceptance, and regulation of related products [50]. The potential risks associated with genetically modified mushroom foods must be addressed through long-term, detailed scientific research and transparent risk assessment.

This article aims to leverage modern molecular biology techniques for precise and efficient breeding in the mushroom industry. It reviews the impact of various high-yield-related genes (genes controlling mycelial growth; genes controlling primordium formation; genes controlling button germination; and genes controlling fruiting body development) on the growth of mushrooms. Additionally, it summarizes current applications of modern molecular biology techniques in mushrooms, exploring a practical technical route for accurately screening high-yield-related genes using transcriptomics, proteomics, and other omics technologies in conjunction with AI. The goal is to cultivate new high-yield mushroom strains through RNAi and gene editing technologies, including CRISPR-Cas9. These findings provide a research foundation and scientific basis for promoting the economic development of the global mushroom industry.

## 2. Genes Controlling Mycelial Growth

### 2.1. Gene Function Characteristics

Mushroom growth occurs in four stages: mycelial, primordium, button, and fruiting body. During the mycelial growth stage, the mycelium consists of numerous interwoven hyphae that form a network structure (Figure 1). The genes regulating mycelial growth primarily facilitate the growth and aggregation of hyphae, which accumulate nutrients for the formation and development of fruiting bodies [51,52,53,54] (Table 1). Key genes such as *Chitinase* (*Chi*), *Glucanase* (*Glu*), *Laccase* (*Lac*), *Hydrophobin* (*Hyd*), and other genes play crucial roles in mycelial growth and aggregation [55,56,57]. *Chi* and *Glu* are vital for the degradation of mycelial cell walls during sexual reproduction, while *Lac* acts as a lignin-degrading enzyme produced by certain fungi, catalyzing the oxidation of phenolic substrates in mycelia [56].

In particular, the *Hydrophobin* gene *Hyd1* plays a significant role in the early stages of fruiting body formation and development of dikaryotic hyphae. Under the stimulation of blue light, the expression of the *Hyd1* gene is significantly upregulated by 2–3 times [56], indicating that illumination at specific wavelengths can promote the expression of genes related to mycelial growth control. It is noteworthy that light is not a necessary environmental condition for mushroom growth, but it can promote certain physiological processes. For example, appropriate levels of scattered light can induce the formation and differentiation of *Pleurotus pulmonarius* primordia, whereas excessive light can lead to abnormal growth of fruiting bodies [58]. Therefore, using light to promote mycelial growth or the expression of genes that regulate mycelial growth should involve actual adjustments based on the photosensitive characteristics of different mushroom varieties.

In the process of mushroom cultivation, temperature is a crucial factor influencing mycelial growth [59]. In particular, the expression of mycelial heat-stress-related genes can either promote or inhibit the growth of mycelia. Because of differential expression of genes such as *Catalase* (*CAT*), *Glutathione peroxidase* (*GPX*), *Heat shock protein* (*HSP*), and *Superoxide dismutase* (*SOD*), under heat stress, mycelial growth significantly increases or decreases, and accelerates the oxidative stress response and energy metabolism in mycelia [60,61,62]. Among them, *Heatshock protein* (*HSP*) and their cognates serve as primary mitigators of cell stress, with significant upregulation observed to alleviate stress-induced cellular damage and protein misfolding, enhancing fungal temperature tolerance [62]. Additionally, the *Metacaspase* (*Mca*) gene regulates physiological activities associated with heat resistance in *Pleurotus ostreatus*, including the positive regulation of mycelial and fruiting body development [63]. RNAi-mediated silencing of the *Mca* gene promotes mycelial growth, resulting in a reduction of approximately 50–65% in *Mca* gene expression within the mycelium [63].

The expression characteristics of heat-stress-related genes in the mycelia of different mushroom varieties indicate that effective regulation of gene expression through the control of external environmental factors such as light and temperature can significantly promote mycelial growth.

**Table 1 jof-10-00767-t001:** Functional characteristics of genes controlling mycelial growth.

Full Name of Gene	Abbreviation	Increasing Production Methods	Function Characteristics
*Chitinase*	*Chi*	Overexpression	Promote the growth of mycelia, enlarge surface area of filamentous fungi cell wall during growth phase [55]
*Glucanase*	*Glu*	Overexpression	Promote the growth of mycelium and the absorption of nutrients, play a major role in cell wall degradation during sexual reproduction [56]
*Hydrophobin*	*Hyd*	Overexpression	Promote the rapid accumulation of mycelia, act in fruiting body initiation and formation of dikaryotic mycelia [56]
*Laccase*	*Lac*	Overexpression	Promote the development of mycelium into a primordium, have important roles for mycelial spread and nutrient uptake during early stage of cultivation [57]
*Catalase*	*CAT*	Overexpression	Promote the formation of dual-core mycelium, performing specialized functions in development and stress resistance [60]
*Glutathione peroxidase*	*GPX*	Overexpression	Promote the accumulation of mycelium, controlling the intracellular H_2_O_2_ content, hyphal branching, antioxidant stress tolerance, cytosolic Ca^2+^ content and ganoderic acid biosynthesis [61]
*Heat shock protein*	*HSP*	Overexpression	Reshape stress-induced cellular functional damage and protein misfolding, control the oxidative stress response and energy metabolism in mycelia [62]
*Superoxide dismutase*	*SOD*	Overexpression	Promote the growth of mycelia, related to the oxidative stress response and energy metabolism in mycelia [62]
*Metacaspase*	*Mca*	Overexpression	Promote the growth of mycelia, enhance mycelial heat stress tolerance [63]
*Phenylalanine ammonia-lyase*	*PAL*	Overexpression	Promote the growth of mycelia, involved in the production of phenolic compounds which perform some functions in the mushroom stipe during the mushroom growth [64]
*Glyceraldehyde-3-phosphate dehydrogenase*	*GPD*	Overexpression	Promote the growth of mycelia, enhance the stress resistance of mushrooms in environmental stress [65]

The genes controlling mycelial growth are expressed not only during the mycelial growth stage, but also during the primordium and fruiting body stage. For example, the expression of the *Phenylalanine ammonia-lyase* (*PAL*) gene in *Flammulina velutipes* gradually increases during the mycelial stage and continues to rise during stipe elongation, with expression levels in the stipe being 10 times higher than those in the cap [64]. Similarly, the transcription levels of *Glyceraldehyde-3-phosphate dehydrogenase* (*GPD*) genes in *Pleurotus ostreatus* are highest during the mycelial stage; as the mushroom matures, both the *PoGPD1* and *PoGPD2* genes are expressed in all parts of the fruiting body, with expression levels in the base being 60–80% higher than in the stipe and cap [65]. These findings highlight the complexity of the gene regulatory network in mushrooms, where genes may be expressed either during specific growth stages or throughout the entire growth process.

### 2.2. Hyd Genes Function Characteristics

Notably, the *Hydrophobin* (*Hyd*) gene has a crucial role in the morphological differentiation of mycelia during development [66]. Overexpressing or inhibiting certain members of the *Hyd* gene family can promote rapid mycelial accumulation within a specific range, thereby shortening primordium formation time and increasing mushroom yield. In a study on the *Hydrophobin* gene *Cmhyd4*, which negatively regulates fruiting body development in edible fungus *Cordyceps militaris*, that deletion of the *Hyd4* gene resulted in a 20–30% increase in fruiting body density [67]. Moreover, the *Hyd* gene, as a key node gene, is expressed in all stages of mushroom growth; for example, 19 *Hyd* genes were screened in the mycelium, primordium, button, and fruiting body of *Grifola frondosa* [68]. This indicates significant potential for breeding high-yield strains by leveraging the expression characteristics of *Hyd* gene family members. The functions of *Hyd* gene family members can be systematically studied through RNAi technology, and then gene editing and CRISPR-Cas9 technology can be utilized to promote the expression of *Hyd* genes, shortening the primordium formation time to increase production. The important characteristics and functions of the *Hyd* gene family in promoting mycelial growth across various mushroom types merit further study.

### 2.3. Future Research Directions

Research using RNAi and gene editing, including CRISPR-Cas9 technology, has demonstrated that genes regulating mycelial growth are promising candidates for breeding. Current technological limitations hinder the accurate screening of genes that promote the formation of these clamp connections, and the research in this area has not been systematic. As a result, it is currently not feasible to precisely and efficiently control the formation of mycelial clamp connections. However, owing to the typical multinucleate nature of mushroom mycelia, which experience two growth stages, namely, the primary and secondary mycelial growth, the formation of secondary mycelia capable of producing fruiting bodies occurs only after the development of clamp connections [69]. The binucleate nature of secondary mycelia complicates the use of existing gene editing methods to introduce stably inherited yield-controlling genes. Further research is necessary on the use of AI technology for accurately screening genes involved in mycelial clamp connection formation. Specifically, research-level AI software could be developed and integrated with existing mushroom genomics databases. The AI would learn from the database’s characteristics and combine data information from professional genetic research websites to facilitate self-learning. Once capable of screening and sequencing genomic information per instructions, the AI could assist researchers in identifying high-yield-related genes within the mushroom gene pool and evaluating their reliability. By combining RNAi and gene editing, including CRISPR-Cas9 technology, it may be possible to stably enhance mycelial growth rates. However, further development of related AI is essential.

## 3. Genes Controlling Primordium Formation

### 3.1. Gene Function Characteristics

The transition from the mycelium to the primordium stage involves a quantitative change, during which mycelia aggregate in large numbers and become twisted together to form primordia [70] (Figure 1). Genes related to aquaporins, proteolysis, mitosis, and lipid and carbohydrate metabolism play significant roles in primordium formation [71] (Table 2).

The aggregation of mycelia and formation of primordia in large quantities prove that promoting the rapid accumulation of mycelia is the key to promoting the formation of primordia. Research has shown that the *Copper sensing transcription factor MAC* (*MAC*) gene of *P. ostreatus* can promote the formation of primordia, with the mycelial growth rate of a *MAC* gene-overexpressing (OE) strain being 23.27–35.83% greater than that of the wild-type strain [72]. The expression level of the *Laccase* (*Lac*) gene in *Volvariella volvacea* peaked during the transition from the mycelium to the primordium stage and then decreased to 20–30% of the peak level during the button stage [73]. The *Hydrophobin* (*Hyd*) gene of *Grifola frondosa* exhibits significant transcript abundance during the mycelium, primordium, and button stages. A remarkably high transcript level of the *Hyd8825* gene was observed during the mycelium stage of *Grifola frondosa* strain CICC^®^50075, approximately 3-fold higher than that at the primordium stage, and the *Hyd9954* gene of the primordia stage was about 380-fold higher than that at the earlier germination stage [69]. This phenomenon indicated that the formation of primordia is positively correlated with the growth and accumulation rate of mycelia. The expression of genes that promote mycelial growth regulates the acceleration of mycelial growth and accumulation, directly or indirectly promoting the formation of primordia. The expression of genes controlling primordium formation is also directly or indirectly regulated by genes that control mycelial growth to some extent. Accelerating mycelial growth and aggregation during the mycelial stage, as well as editing genes to control primordium formation, can effectively promote primordium formation.

**Table 2 jof-10-00767-t002:** Functional characteristics of genes controlling primordium formation.

Full Name of Gene	Abbreviation	Increasing Production Methods	Function Characteristics
*Hydrophobin*	*Hyd*	Overexpression	Promote the formation of primordia, assist the filamentous fungi to fulfill life activities [68]
*Copper sensing transcription factor MAC*	*MAC*	Overexpression	Promote the formation of primordia, modulating the growth, development and thermotolerance [71]
*Laccase*	*Lac*	Overexpression	Promote the formation of primordia and formation of Spores [73]

### 3.2. Future Research Directions

By combining omics and AI technologies, we can comprehensively study the regulatory network between genes controlling mycelial growth and those regulating primordium formation. Using gene editing technology to overexpress key high-yield-related genes can promote the rapid formation of primordia. Additionally, it is necessary to study the effects of environmental factors on genes controlling primordium formation and to explore new methods for promoting primordium formation via RNAi and gene editing, including via CRISPR-Cas9 technology.

## 4. Genes Controlling Button Germination

### 4.1. Gene Function Characteristics

After the formation of the primordium, it gradually differentiates into a button with a diameter of 3–5 mm. The number of germinating buttons is crucial for the formation of numerous fruiting bodies and significantly affects fruiting body yield [74] (Figure 1) (Table 3). In a study of *Ganoderma lucidum*, it was found that during the development from the button to the fruiting body, the expression level of *Cytochrome P450s* (*P450s*) genes gradually increased, reaching levels 20 times higher at the button stage compared to the mycelial stage. Following sporulation, the expression level of these genes gradually decreased [75], indicating that *Cytochrome P450s* genes play a key role in the formation of the fruiting body and that promoting the expression of these genes may benefit button germination. However, the mechanisms regulating *Cytochrome P450s* gene expression during button germination are still unclear and require further research. Genes such as *Lysine methyltransferase* (*Lysine*), *L-lysine 6-monooxygenase* (*L-lysine 6*), and *Fasciclin-like protein 1* (*Flp*) significantly impact the development of the button stage. Under the stimulation of red and blue light, the overexpression of *Lysine* and *L-lysine 6* genes promotes the synthesis of lysine-related substances, indirectly increasing the density of buttons and the yield of fruiting bodies by 1–2 times. In the RT-PCR assay, the *Flp* gene exhibits the clearest band during the button stage; compared to the primordium stage, the *Flp* gene specifically transcribes in large quantities during the button stage [76,77]. This indicates that environmental factors such as light are crucial for inducing the differentiation of the primordia to buttons. Notably, in the study of the *Metacaspase* gene *PoMCA1*, which enhances the mycelial heat stress tolerance and regulates the development of fruiting bodies in *Pleurotus ostreatus*, it was found that interference with the *PoMCA1* gene resulted in a 42.00–57.00% increase in the density of mushroom buds and young fruiting bodies [59]. This indicates that the expression of some genes controlling button germination may play an inhibitory role in the process, suggesting that these genes could serve as key targets for CRISPR-Cas9 breeding.

### 4.2. Future Research Directions

At present, although many genes are known to control button germination, the existing research mainly categorizes genes related to button germination and those involved in fruiting body development as high-yield-related genes. However, there are significant differences in the expression of these high-yield-related genes during the button stage and the young fruiting body stage. Taking *Pleurotus ostreatus* as an example, during the development process from the button to the fruiting body stage, the proportion of highly expressed genes reaches 47.20% [74]. Therefore, the genes controlling button germination should be studied separately and systematically to identify ways to promote mushroom bud germination using molecular biology techniques. After conducting in-depth research on the genes controlling button germination, active substances that can stimulate the overexpression of these genes could be incorporated into the mushroom package to effectively promote button germination. However, further research is needed to identify which substances and methods can achieve low-cost stimulation of this process.

## 5. Genes Controlling Fruiting Body Development

### 5.1. Gene Function Characteristics

The button germinates into a young fruiting body, which gradually develops into characteristic structures. The fruiting body developmental stage determines the final yield of mushrooms, and significant upregulation of gene expression controlling fruiting body development occurs during this stage [78,79] (Figure 1) (Table 4). For example, the expression level of the *Heat shock protein* (*HSP*) gene in Grifola frondosa increased 5–6-fold during the differentiation of fruiting bodies, indicating that the *HSP* gene promotes this differentiation [80]. Additionally, the *Laccase* gene in *V. volvacea* is strongly expressed during the fruiting body development stage, with peak gene expression fluctuating between 21% and 42% during the button stage of fruiting body emergence [81]. The expression of genes controlling fruiting body development varies in different parts of the young fruiting body and may also be present throughout the entire fruiting body [82]. For example, the transcription level of *TmPAL1*, a member of the *Phenylalanine ammonia-lyase* (*PAL*) gene family in *Tricholoma matsutake*, is higher in the cap, stipe, and base of mature fruiting bodies, whereas the transcription level of *TmPAL2* is elevated in the gills, exceeding that in the cap by more than 1.7 times [83]. The *Cys2His2 zinc finger protein* (*C2h2*) gene in *A*. *bisporus* is highly expressed at different stages of fruiting body development, with expression levels more than four-fold greater in the cap of stage I buttons, the gill of stage II buttons, and the veil tissue of young fruiting bodies [84]. These findings indicate that genes may exhibit differential expression at different positions during the development of the fruiting body.

Research has revealed that high-yield-related genes, which are highly expressed during the mycelia stage, also exhibit high expression during the fruiting body stage, greatly influencing the final yield of mushroom fruiting bodies. For example, the transcription level of the *Gβ-like protein CPC-2* (*CPC2*) gene in the primordia of *F. velutipes* is 5.71 times greater than that in the mycelia, and overexpression of the *CPC2* gene can shorten the cultivation time of *F. velutipes* by 7.3% and increase the height of the stipe by 14.8% [85]. This finding also demonstrates the phenomenon of phased expression of high-yield-related genes. This suggests that based on the characteristics of gene expression at different stages, molecular biology techniques can be used to promote the expression of genes controlling mycelial and primordium development in advance, thereby promoting the growth of fruiting bodies and achieving the goal of increased production.

### 5.2. Future Research Directions

Although modifying high-yield-related genes during the fruiting body stage is challenging, it is possible to screen for genes that control fruiting body development with significant application value using omics and AI technology. High-yield strains can be subjected to tissue culture to obtain mycelia. Through gene editing techniques, including CRISPR-Cas9, genes promoting fruiting body development can be overexpressed, or the expression of genes that inhibit this development can be suppressed during the mycelia stage. This approach can enhance the growth of fruiting bodies and increase the final yield. However, further research is needed to determine whether the genes controlling fruiting body development can be stably transmitted from the mycelial stage to the fruiting body stage.

## 6. Conclusions

Research on high-yield-related genes in mushrooms (genes controlling mycelial growth, genes controlling primordium formation, genes controlling button germination, and genes controlling fruiting body development) can effectively enhance the economic development of the mushroom industry. In the future, it will be possible to comprehensively combine advanced scientific research-level AI and omics technologies. By leveraging the data characteristics of the omics-related databases, we can integrate research-level AI to accurately screen for high-yield-related genes in mushrooms, focusing particularly on those with high application value. To effectively use high-yield genes to promote mushroom growth, specific applications must be based on the expression characteristics of different high-yield genes. Combining RNAi to verify gene functions with gene editing techniques, such as CRISPR-Cas9, can facilitate the cultivate new mushroom strains. This approach provides a theoretical foundation for researching high-yield-related genes in mushrooms and enhancing production, ultimately contributing to the economic development of the mushroom industry.

## Figures and Tables

**Figure 1 jof-10-00767-f001:**
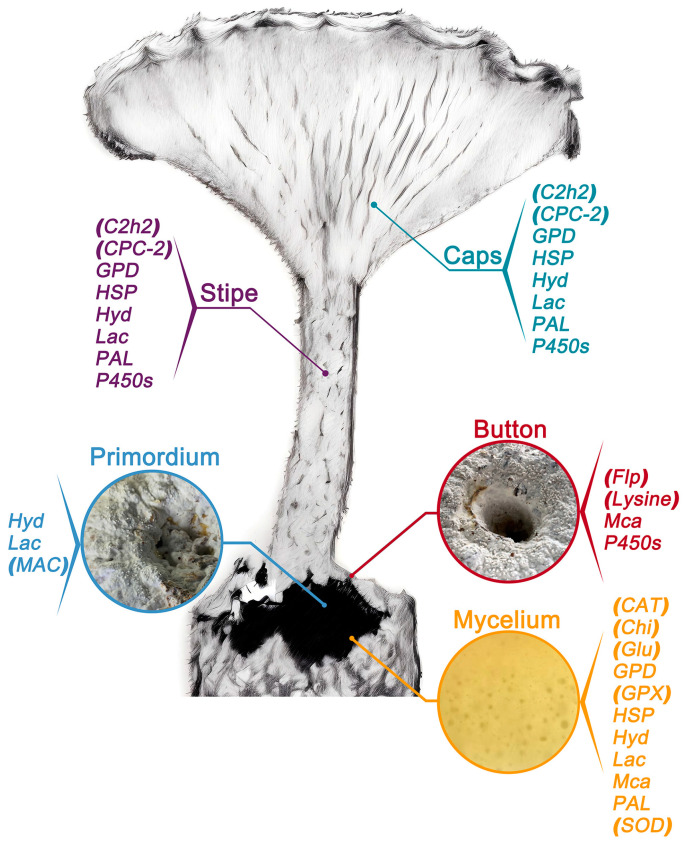
Distribution of yield-related gene expression at different stages of mushroom growth, including the mycelium, primordium, button, and fruiting body stages. Genes expressed at specific stages are labeled in parentheses. *Catalase* (*CAT*); *Chitinases* (*Chi*); *Cytochrome P450s* (*P450s*); *Cys2His2 zinc finger protein* (*C2h2*); *Fasciclin-like protein 1* (*Flp*); *Gβ-like protein CPC-2* (*CPC-2*); *Glucanases* (*Glu*); *Glyceraldehyde-3-phosphate dehydrogenase* (*GPD*); *Glutathione peroxidase* (*GPX*); *Heat shock protein* (*HSP*); *Hydrophobin* (*Hyd*); *Copper-sensing transcription factor* (*MAC*); *Laccase* (*Lac*); *Lysine methyltransferase* (*Lysine*); *Metacaspase* (*Mca*); *Phenylalanine ammonia-lyase* (*PAL*); *Superoxide dismutase* (*SOD*).

**Table 3 jof-10-00767-t003:** Functional characteristics of genes controlling button germination.

Full Name of Gene	Abbreviation	Increasing Production Methods	Function Characteristics
*Fasciclin-like protein 1*	*Flp*	Overexpression	Promote the formation of button, plays a role in cellular differentiation and development in ubiquitous tissues [77]
*Lysine methyltransferase*	*Lysine*	Overexpression	Promote the formation of button, controlled accumulation of lysine in fruiting body [76]
*Metacaspase*	*Mca*	Suppressive expressed	Increase the density of button [63]
*Cytochrome P450s*	*P450s*	Overexpression	Promote the development of button into young fruiting bodies, play a central role in fruiting body development [75]

**Table 4 jof-10-00767-t004:** Functional characteristics of genes controlling fruiting body development.

Full Name of Gene	Abbreviation	Increasing Production Methods	Function Characteristics
*Cys2His2 zinc finger protein*	*C2h2*	Overexpression	Promote the development of fruiting body, play a role in control of outgrowth of primordia into fruiting bodies or to play a role in expansion of the fruiting body [84]
*Gβ-like protein CPC-2*	*CPC-2*	Overexpression	Promote the development of fruiting body [85]
*Glyceraldehyde-3-phosphate dehydrogenase*	*GPD*	Overexpression	Promote the formation of fruiting body, promote the the formation and expansion of mycelial branches [65]
*Heat shock protein*	*HSP*	Overexpression	Promote the differentiation of fruiting body structure [80]
*Hydrophobin*	*Hyd*	Overexpression	Promote the formation of fruiting body [67]
*Laccase*	*Lac*	Overexpression	Promote the formation of fruiting body [81]
*Phenylalanine ammonia-lyase*	*PAL*	Overexpression	Promote the development of fruiting body [83]
*Cytochrome P450s*	*P450s*	Overexpression	Promote the formation of fruiting body [75]

## Data Availability

Not applicable.

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
