# Peer review of "High-Yield-Related Genes Participate in Mushroom Production"

_jof, 2024, doi:10.3390/jof10110767_

Round 1
Reviewer 1 Report
The mushrooms industry is indeed growing fast hand by hand with molecular genetics technology. I agree with the authors that the current and probably more to come theologies can benefit this industry.
The authors review a number of genes that could be involved in the distinguished developmental stages of various mushrooms. This information is interesting and useful for mushrooms future research.
The authors put all mushrooms in one basket, I wonder if findings for one mushroom are valid for all, can they discuss this point. For example, fruiting body growth in P. ostreatus requires light while A. busporus not. As far as I know different genes must be involved in different mushrooms.
Is it suggested to move genes from one species to another to obtain high yield – I suggest to discuss this possibility and also the current attitude to transgenic mushroom, is it acceptable globally.
The authors discuss high yield improvement only – I suggest to mention in short quality characteristics such as nutritional and medicinal values that can be modified genetically.
Specific comments
The term " button germination" is not clear to me and I was confused with spore germination. Can the authors use a different term?
Can the authors explain in more details how AI can help
Reference 43,33 line 217 – Ganoderma should be in italics
Line 234, 250 reference 21 Pleurotus should be in italics
Author Response
Dear Editor and Reviewer,
Thank you for your valuable comments on our article. According to your suggestions, we have corrected several points addressed by reviewers in our previous manuscript and supplemented several data. Therefore, extensive revisions were made and we believed that the quality of the manuscript has been greatly improved. The detailed point-by-point responses were listed below.
Reviewer: 1
The mushrooms industry is indeed growing fast hand by hand with molecular genetics technology. I agree with the authors that the current and probably more to come theologies can benefit this industry.
My Response: Thank you for recognizing the core viewpoint of this article.
The authors review a number of genes that could be involved in the distinguished developmental stages of various mushrooms. This information is interesting and useful for mushrooms future research.
My Response: Thank you for your recognition of the research on high-yield genes related to mushrooms.
The authors put all mushrooms in one basket, I wonder if findings for one mushroom are valid for all, can they discuss this point. For example, fruiting body growth in P. ostreatus requires light while A. busporus not. As far as I know different genes must be involved in different mushrooms.
My Response: Relevant content has been supplemented for further explanation. Modified and supplemented sentences:
It is noteworthy that light is not a necessary environmental condition for mushrooom growth, but it can promote certain physiological processes. For example, appropriate levels of scattered light can induce the formation and differentiation of Pleurotus pulmonarius primordia, whereas excessive light can lead to abnormal growth of fruiting bodies [59]. Therefore, using light to promote mycelial growth or the expression of genes that regulate mycelial growth should involve actual adjustments based on the photosensitive characteristics of different mushroom varieties (Line 125-131; Line 490-492).
To effectively use high-yield genes to promote mushroom growth, specific applications must be based on the expression characteristics of different high-yield genes. (Line 343-345).
Is it suggested to move genes from one species to another to obtain high yield – I suggest to discuss this possibility and also the current attitude to transgenic mushroom, is it acceptable globally.
My Response: The advantages and disadvantage of gene editing mushroom were discussed, and application feasibility be discussed.
Supplementary sentences:
Although molecular biology technology has significantly shortened the breeding cycle of mushrooms and accelerated the research and development of new varieties, challenges remain regarding safety, social acceptance, and regulation of related products [51]. The potential risks associated with genetically modified mushroom foods must be addressed through long-term, detailed scientific research and transparent risk assessment (Line 92-97).
The authors discuss high yield improvement only – I suggest to mention in short quality characteristics such as nutritional and medicinal values that can be modified genetically.
My Response: Relevant content has been added to the article introduction.
Supplementary sentences:
Taking the research on multiplex gene precise editing and large DNA fragment dele-tion using the CRISPR-Cas9-TRAMA system in edible mushroom Cordyceps militaris as an example, the synthetases of cordycepin and ergothioneine possess functional char-acteristics that inhibit cancer and skin aging. Employing CRISPR-Cas9 technology to investigate the functional characteristics of the cordycepin and ergothioneine genes, while simultaneously enhancing the yield of Cordyceps militaris, could significantly promote the economic benefits for both the pharmaceutical and cosmetic industries [46]. (Line 82-89; Line 462-464).
Specific comments
The term " button germination" is not clear to me and I was confused with spore germination. Can the authors use a different term?
My Response: Button refers to the immature fruiting body, which is the transitional stage of primordium development into mature fruiting body. Spores germinate and develop into hyphae, while button is the stage of fruiting body development. Generally, button refers to the stage of young fruiting body.
Can the authors explain in more details how AI can help
My Response: AI related content has been supplemented in the article. Line 202-207 explains how to combine AI with molecular biology for research.
Supplementary sentences:
The rapid processing, analysis, and precise targeting capabilities of AI make it particularly well-suited for the screening of high-yield genes [34-38]. Currently, the fusion of biology and artificial intelligence has been extensively applied in the field of syn-thetic biology, resulting in the establishment of relevant technology platforms and databases. However, the integration of molecular biology and artificial intelligence still requires further investigation and research for effective application [39]. (Line 66-71; Line 445-446).
Reference 43,33 line 217 – Ganoderma should be in italics
My Response: Ganoderma has been modified to italic (Line 253; Line 431; Line 454).
Line 234, 250 reference 21 Pleurotus should be in italics
My Response: Pleurotus ostreatus has been modified to italic (Line 270; Line 281; Line 407).
Reviewer 2 Report
The authors have prepared a review of literature data on an important scientific problem - the study of edible mushroom genes, the expression of which regulates their yield. However, the analysis of the collected data needs to be significantly improved.
Main remarks
1. It is necessary to distinguish between genes whose expression needs to be increased or decreased to create high-yielding strains. When citing articles, this important information is completely missing, although it is given in the articles. These facts determine the choice of methods for increasing yield.
2. The authors consider the stages of mycelial growth, formation of primordia, buttons, and fruiting bodies separately. It is necessary to distinguish between target genes that work at all stages of ontogenesis and genes that are specific to each stage.
3. It is necessary to exclude multiple references to artificial intelligence, iRNA, and CRISPR-Cas9 methods. The manuscript does not provide articles using them, and there are no specific proposals from the authors for their use.
4. There is no information on what molecular genetic methods have already been used to obtain high-yielding strains and what the success of these studies is.
Minor remarks
1. The manuscript contains many imprecise and unfortunate expressions. For example, "...Lysine and L-lysine 6 genes increased the density of button and the yield of fruiting bodies …" (P 227-228). It is not the genes that increase the density. "Moreover, due to active cell differentiation and metabolism during the germination stage, there are significant fluctuations in the expression of various gene families, making genetic modification more difficult." (248-250). What is "germination stage" and what gene families are we talking about? "The expression of genes controlling fruiting body development varies in different parts of the mature fruiting body …" (P 268-269). Are the genes controlling fruiting body development expressed if the fruiting body has already reached maturity? "...during the mycelial primordium stage …" (280-281). Such a stage does not exist.
2. The authors are not sufficiently familiar with the morphology and physiology of edible mushrooms.
- Figure 1. Primordia cannot be located under the mycelium.
- C 260 "characteristic structures such as the cap, gills, and stipe". Not all species of edible mush rooms have a cap and a stipe. If a cap exists, the hymenophore is part of it.
- C 284 "gill tissue". Mushrooms do not have tissue.
- C 231-232. The role of light in inducing fruiting was known long before genetic studies.
3. The English language needs improvement.

Author Response
Dear Editor and Reviewer,
Thank you for your valuable comments on our article. According to your suggestions, we have corrected several points addressed by reviewers in our previous manuscript and supplemented several data. Therefore, extensive revisions were made and we believed that the quality of the manuscript has been greatly improved. The detailed point-by-point responses were listed below.
Reviewer: 2
- It is necessary to distinguish between genes whose expression needs to be increased or decreased to create high-yielding strains. When citing articles, this important information is completely missing, although it is given in the articles. These facts determine the choice of methods for increasing yield.
My Response: Relevant contents have been added to the table that explain how to use high-yield genes to increase yield (Line 158; Line 219; Line 275; Line 312).
- The authors consider the stages of mycelial growth, formation of primordia, buttons, and fruiting bodies separately. It is necessary to distinguish between target genes that work at all stages of ontogenesis and genes that are specific to each stage.
My Response: According to the reviewer request, specific stages expressed genes have been labeled in Figure 1 (Line 150).
- It is necessary to exclude multiple references to artificial intelligence, iRNA, and CRISPR-Cas9 methods. The manuscript does not provide articles using them, and there are no specific proposals from the authors for their use.
My Response: AI related content has been supplemented in the article, and artificial intelligence, RNAi, and CRISPR-Cas9 content was adjusted. Line 173-186 explains how to combine AI with molecular biology for research.
Supplementary sentences:
Modern molecular biology techniques, including omics, RNA interference (RNAi), and gene editing- particularly the novel CRISPR-Cas 9 technology—enable precise identification and editing of target genes, facilitating the targeted enhancement of high-yield characteristics in mushrooms [29-33]. Through omics research, it is possible to identify target genes that regulate high-yield characteristics in mushrooms. This can be followed by precise screening of high-yield-related genes using advanced artificial intelligence (AI) technologies. The rapid processing, analysis, and precise targeting capabilities of AI make it particularly well-suited for the screening of high-yield genes [34-38]. Currently, the fusion of biology and artificial intelligence has been extensively applied in the field of synthetic biology, resulting in the establishment of relevant technology platforms and databases. However, the integration of molecular biology and artificial intelligence still requires further investigation and research for effective application [39]. (Line 60-71; Line 445-446).
After screening the high-yield-related genes, RNAi technology can be employed to systematically investigate the function of the target high-yield-related genes. Its technical advantages include rapid action, high stability, relative ease of operation, and the capability to target multiple genes simultaneously in a single procedure [40-42] (Line 72-75).
- There is no information on what molecular genetic methods have already been used to obtain high-yielding strains and what the success of these studies is.
My Response: According to the reviewer comments, literature has been added to improve the relevant content in the article.
Supplementary sentences:
Currently, the use of molecular biology markers to support hybrid breeding technology offers distinct advantages traditional breeding methods. For example, the new cultivar Agaricus bisporus cultivar Fumo 48, under optimal cultivation conditions, demonstrates average yields comparable to the prominent foreign cultivar A15, while exceeding those of As2796 by 20%–25%.This cultivar is well-suited for year-round modern farm cultivation and appropriate for fresh market sales [28]. Nevertheless, the broader promotion and application of other molecular biology techniques in mushroom breeding necessitate the establishment of a mature and secure technical framework (Line 51-59).
Minor remarks
- The manuscript contains many imprecise and unfortunate expressions. For example, "...Lysine and L-lysine 6 genes increased the density of button and the yield of fruiting bodies …" (P 227-228). It is not the genes that increase the density. "Moreover, due to active cell differentiation and metabolism during the germination stage, there are significant fluctuations in the expression of various gene families, making genetic modification more difficult." (248-250). What is "germination stage" and what gene families are we talking about? "The expression of genes controlling fruiting body development varies in different parts of the mature fruiting body …" (P 268-269). Are the genes controlling fruiting body development expressed if the fruiting body has already reached maturity? "...during the mycelial primordium stage …" (280-281). Such a stage does not exist.
My Response: Revised according to the reviewer comments.
Modified sentence:
Under the stimulation of red and blue light, the overexpression of Lysine and L-lysine 6 genes promotes the synthesis of lysine related substances, indirectly causing the density of buttons and the yield of fruiting bodies by 1-2 times; in the RT-PCR assay, the Flp gene exhibits the clearest band during the button stage, compared to the primordium stage, the Flp gene specifically transcribes in large quantities during the button stage [77, 78] (Line 262-267).
The expression of genes controlling fruiting body development varies in different parts of the young fruiting body and may also be present throughout the entire fruiting body [83] (Line 300-302).
Research has revealed that high-yield-related genes, which are highly expression during the mycelia stage also exhibit high expression during the fruiting body stage, greatly influencing the final yield of mushroom fruiting bodies (Line 313-315).
- The authors are not sufficiently familiar with the morphology and physiology of edible mushrooms.
My Response: Thank you for the valuable comments provided by the reviewer. The revisions have been made according to the requirements.
- Figure 1. Primordia cannot be located under the mycelium.
My Response: The order of primordia and mycelium in the image has been modified as required (Line 150).
- C 260 "characteristic structures such as the cap, gills, and stipe". Not all species of edible mush rooms have a cap and a stipe. If a cap exists, the hymenophore is part of it.
My Response: This part of the article has been modified.
Modified sentence:
The button germinates into a young fruiting body, which gradually develops into characteristic structures (Line 292-293).
- C 284 "gill tissue". Mushrooms do not have tissue.
My Response: The relevant content has been modified.
Modified sentence:
The Cys2His2 zinc finger protein gene (C2h2) gene in A. bisporus is highly expressed at different stages of fruiting body development, with expression levels more than four-fold greater in the cap of stage I buttons, the gill of stage II buttons, and the veil tissue of young fruiting bodies [85] (Line 306-309).
- C 231-232. The role of light in inducing fruiting was known long before genetic studies.
My Response: We have made modifications to the relevant content and added information on the effect of light on mushroom growth in Line 121-131 (Line 121-131).
Supplementary sentences:
In particular, the Hydrophobin gene Hyd1 plays a significant role in the early stages of fruiting body formation and development of dikaryotic hyphae. Under the stimula-tion of blue light, the expression of Hyd1 gene is significantly upregulated by 2-3 times [57]. Indicating that illumination at specific wavelengths can promote the expression of genes related to mycelial growth control. It is noteworthy that light is not a necessary environmental condition for mushrooom growth, but it can promote certain physio-logical processes. For example, appropriate levels of scattered light can induce the formation and differentiation of Pleurotus pulmonarius primordia, whereas excessive light can lead to abnormal growth of fruiting bodies [59]. Therefore, using light to promote mycelial growth or the expression of genes that regulate mycelial growth should involve actual adjustments based on the photosensitive characteristics of different mushroom varie-ties (Line 121-131).
- The English language needs improvement.
My Response: Thank you for the valuable comments provided by the reviewer. We have carefully revised the English expression in the article.